# Generation of PVP Membranes Using Extracts/Phenolic Fraction of *Dysphania ambrosioides*, *Opuntia ficus-indica*, and *Tradescantia pallida*

**DOI:** 10.3390/polym15244720

**Published:** 2023-12-15

**Authors:** Orlando Zaca Moran, Wendy Argelia García Suastegui, Jonathan Hillel Cruz San Juan, Lawrence Christopher López Méndez, Valentin López Gayou

**Affiliations:** 1Laboratorio de Bionanotecnología, Centro de Investigación en Biotecnología Aplicada, Instituto Politécnico Nacional (IPN-CIBA), Tlaxcala 90700, Mexico; ozacam@ipn.mx (O.Z.M.); jcruzs2100@alumno.ipn.mx (J.H.C.S.J.); 2Instituto de Ciencias, Benemérita Universidad Autónoma de Puebla, Puebla 72570, Mexico; wendy.garcias@correo.buap.mx; 3Independent Researcher, Puebla 72000, Mexico; l.christopherl.0211@gmail.com

**Keywords:** electrospun membranes, therapeutic membranes, phenol compounds, ATR_FTIR, microplastics

## Abstract

In the present work, electrospun membranes of polyvinylpyrrolidone (PVP) nanofibers were manufactured using extracts and phenolic fractions of *Dysphania ambrosioides* (epazote), *Opuntia ficus-indica* (nopal), and *Tradescantia pallida* (chicken grass). The characterization of the membranes was carried out by scanning electron microscopy and Fourier transform infrared spectroscopy. The membranes synthesized through the use of the extracts generally showed a slight decrease in the diameter of the fibers but an increase in the size of the pores due to the presence of nanoparticles (rosaries) on the surface of the fibers, while the membranes synthesized using the phenolic fraction demonstrated an inversely proportional relationship between the compounds of this family with the diameter of the fibers and the size of the pore, allowing to elucidate part of the polymerization mechanisms of PVP nanofibers, in addition to proposing a reaction mechanism in the interaction between PVP and phenolic compounds for surface functionalization. Likewise, we demonstrate that the generation of reaction seeds through functionalization allows the addition of other compounds to the fibers in the membranes synthesized using the complete extract.

## 1. Introduction

Skin wounds have been a prevalent health problem since the beginning of the human species; however, the ways in which they have been addressed have changed little. There are many studies that affirm that “traditional” healing methods are useless, including increasing the probability of contracting some type of infection, delaying the healing process, and causing discomfort and more pain to the patient [1,2,3]. In recent decades, active healing devices have been developed, such as hydrogels, foams, hydrocolloids, and ointments, which have helped promote the healing process [4,5]. However, one of their greatest disadvantages is that they are occlusive methods that do not allow gas exchange and natural perspiration of the wound, producing maceration of the tissue [6]. That is why various work groups around the world seek to overcome the challenges presented by the wound healing process by proposing new technologies and innovative devices that intervene actively and dynamically with the different elements and events that occur in the healing process [7,8,9].

Electrospinning is a promising method for the rapid and cost-effective production of nanofibers (NFs) from a wide variety of natural, synthetic, and hybrid polymers. This method of manufacturing NFs is composed of three basic elements: (1) a syringe with a metal needle to transport the polymer solution with a controlled feeding rate; (2) a high-voltage generator; and (3) a metal collector in which to deposit the final fibers. In a stable electrospinning process, the high voltage generates repulsive forces in the liquid polymer to overcome its surface tension at the tip of the needle, forming what is known as a Taylor cone. Subsequently, a jet of liquid is formed that travels through the air and is directed towards the collector plate, which is also attached to the voltage generator. Before the deposition of the solid fibers on the collector, the solvents evaporate on the way from the needle to the collector, and, finally, the fibers manage to reach the plate completely dry to form a membrane based on NFs [10]. The deposition of these NFs forms nanostructured membranes with a high surface/volume ratio and high porosity, and that also have the advantage of being able to introduce different therapeutic compounds. The molecules can be embedded directly into the polymer matrix or attached to the fiber surface. Given their properties, NFs have opened the doors to designing a new generation of medical textiles [7]. The versatility of the electrospinning technique allows the electrospun membranes to mimic the extracellular environment while being able to administer several therapeutic agents in a focused manner that can also act for a long time. Additionally, because NFs are made from biocompatible polymers, they do not require removal and allow the beneficial effects to act constantly. Unlike the simple electrospinning system (two components: a polymer dissolved in a solvent), the integration of other agents in the spinning solution generates other types of physical–molecular interactions that provide greater resistance and stability and allow good electrospinning. The main forces that can be found in the polymer–extract interaction are the polymer’s own bonds and hydrogen bonds. These combined interactions increase the viscosity of the solution, stabilizing the jet and providing the fibers with greater stability [10,11,12,13,14,15,16].

Around the world, the therapeutic potential of plants has been known since ancient times, many of which have been studied for the characterization of their biologically active compounds. However, many other compounds present different activities that can mediate other processes, from enzymatic to structural. In recent years, there has been great research interest in the encapsulation of bioactive plant extracts in electrospun materials. Different authors have reported the incorporation of different plant extracts in NFs, such as mucilage from *Opuntia ficus-indica* [17,18], *centella asiatica extract* [19], *Hypericum perforatum extract* [20], *Annona muricata leaf extract* [21], *Calendula officinalis extract* [22], *Lawsonia inermis extract* [23], *Moringa oleifera extract* [24], *Garcinia mangostana extract* [25], *Matricaria chamomilla extract* [26], and *Curcuma longa extract* [27], to mention a few.

*Dysphania ambrosioides* (DA), popularly known as epazote, is an aromatic perennial plant that is used as a condiment and as a medicinal plant in Mexico and many other Latin American countries, about which there have been reports of biological activities such as amoebicidal, anticancer, antibacterial, antidiabetic, antidiarrheal, antifungal, anti-inflammatory, antinociceptive, antioxidant, antiulcer, anxiolytic, immunomodulatory, and vasorelaxant [28,29,30]. *Opuntia ficus-indica*, commonly known as nopal, is a shrub species of the genus Opuntia of the cactus family, widely consumed in Mexico, which has been documented with anti-inflammatory, hypoglycemic, stomach ulceration inhibitory, neuroprotective, and antioxidant properties. It has also been used to treat diabetes, burns, bronchial problems, asthma, and indigestion [31,32]. *Tradescantia pallida* is a species of herbaceous and perennial plant endemic to Mexico, used mainly ornamentally; however, it has been used in the biomonitoring of contamination through the frequency of micronuclei (somatic mutations induced by carcinogens), in addition to having antioxidant, α-glucosidase inhibitory, antimicrobial, and hepatoprotective activity [33,34,35,36].

In this work, PVP membranes were manufactured by electrospinning using extracts and fractions obtained from the three plants mentioned above. The aim of the study reported here was to manufacture and characterize membranes based on a biocompatible polymer (PVP) containing an extract/phenol fraction of the plants obtained by electrospinning. However, the results obtained allowed not only the determination of the morphology and structure obtained by their manufacturing processes but also the elucidation of parts of the reactions that are occurring for the functionalization of the fibers for future applications.

## 2. Materials and Methods

### 2.1. Obtaining Biological Material

The plant material of DA and OFI were obtained in a local market in the community of San Andrés Calpan, located in the state of Puebla, México, and the plant material of TP was obtained from gardens in the northern mountains of Puebla.

### 2.2. Obtaining the Extracts

For the preparation of the DA and TP extracts, stems and leaves were used, while for the OFI extract, the chlorophyllic parenchyma of mature cladodes was used. The plant material was dehydrated in the shade at room temperature and subsequently pulverized. To obtain the hydroalcoholic extract, 40 g of the ground dry plant material was added to 800 mL of methanol:water at a 1:1 (*v*/*v*) ratio, taking it to 75 °C for one hour, then the supernatant was removed, and the procedure was repeated 6 times. The extract was concentrated via low-pressure distillation using a rotary evaporator (Buchi 490, Flawil, Switzerland) and subsequently stored at 4 °C.

### 2.3. Phenolic Fraction

To obtain the phenolic fraction, 0.5 g of extract from each plant was dissolved in 20 mL of 80% methanol (Meyer, Ciudad de México, Mexico) with 20 mL of distilled water, macerating and stirring for an hour. The mixture was allowed to precipitate for one hour, and the supernatant was removed and saved for subsequent steps. To the remaining solids, 20 mL of 70% acetone and 20 mL of distilled water were added, repeating the previous extraction process. Both supernatants were mixed and refrigerated at 4 °C for one day. The next day, the mixtures of both supernatants were filtered to obtain a crystalline liquid phenolic fraction. These were concentrated via low-pressure distillation using a rotary evaporator (Buchi 490, Flawil, Switzerland) and subsequently resuspended in 10 mL of ethanol and stored refrigerated at 4 °C until use.

### 2.4. Preparation of the Membranes

PVP was purchased from Sigma-Aldrich,(Merck KGaA, Darmstadt, Germany) with an average molecular weight of 360,000 g/mol (cat PVP360). The acetone (cat 0025), isopropyl alcohol (cat 0405), and methanol (cat 5420) of analytical grade were purchased from Meyer (México), and distilled water was obtained from a model 90750 water/pro/RO purifier (Labconco, Kansas City, KS, USA).

Several types of nanostructured membranes were prepared: PVP membranes, PVP membranes made with the complete extracts, and PVP membranes made with the phenolic fractions of each plant. These membranes were manufactured with a double-syringe infuser (KD Scientific, Holliston, MA, USA), a high-voltage source with an operating range of 1–30 kV (Spellman, Bronx, NY, USA), and a circular copper collector plate that was covered with aluminum foil for each membrane made. The parameters for manufacturing were humidity and ambient temperature, with a current of 15 kV, a flow of 0.5 mL/h, a working distance of 13 cm between dispenser and collector, and a deposition time of 10 min. For the membranes manufactured with phenolic extract or fraction, concentrations of 7% wt. of polymer and 5% wt. of extracts or phenolic fraction were used. The DA and OFI extracts, as well as the phenolic fractions, were dissolved in isopropyl alcohol; only the TP extract was dissolved in methanol to improve its solubility. All labels used for membranes within the article are found in Table 1.

### 2.5. Characterization of Membranes

The spectroscopic characterization of the extract and membranes was carried out with a VERTEX 70 FTIR spectrometer (BRUKER Optics, Ettlingen, Germany) using the attenuated total reflectance (ATR) technique at a resolution of ±4 cm^−1^. The morphological characterization of the microstructures was carried out using a VEGA TS 5136SB microscope (TESCAN, Brno-Kohoutovice, Czech Republic) with a high vacuum resolution (0.009 Pa) of 3.5 nm, secondary (SE) and backscattered (BSE) detectors, and a magnification range of 4–500,000× and an accelerating voltage of 0.5–30 kV. The analysis of the images was carried out using the image J program, taking 25 measurements per region of interest, for a total of 100 measurements per image. A statistical analysis was carried out using descriptive statistics.

## 3. Results

### 3.1. Morphological Characterization of Membranes

In general, PVP membrane was used as a control. Figure 1 shows the membranes obtained under different conditions, other representative images at different scales can be seen in the Appendix A. In the case of the PVP membrane, random but homogeneous-diameter nanofibers (NFs) were observed without defects and with a uniform morphology. However, in the case of the membranes made with the complete extracts, extract agglomerates of variable size, like rosaries, were present in the 3 types of membranes. The membranes made with the phenolic fractions did not show such structures between the NFs on the lattice.

PVP membranes showed the largest average diameter of NFs, with 915.17 nm, while the membranes made with the extracts showed average diameters of NFs, with the average diameter from DA + PVP of 544.04 nm, OFI + PVP of 680.5 nm, and TP + PVP of 572.57 nm; showing an average diameter decrease of approximately 34%. In the same way, the membranes made with the phenolic fractions showed the smallest average diameters of NFs, with that from FR-DA + PVP of 323 nm, that from FR-OFI + PVP of 338.21 nm, and that from FR-TP + PVP of 331.48 nm, with a reduction in the average diameter of 64% compared to the control, as can be seen in Figure 2.

It has been reported that the addition of different plant extracts, such as *C. asiatica*, *C. longa*, *L. inermis*, and *M. chamomilla*, to different polymers, such as PCL, chitosan/PEO, chitosan/PVA, and polylactic acid (PLA), reduces the diameter of the fiber from 10 to 50% [18,22,25,26]. This is mainly attributed to the modification of the viscosity of the solution by adding new compounds to the mixture, for which the fractionation of the extracts for the identification of the compounds present in them is necessary.

Regarding the average pore area of the membrane, PVP membranes showed an area of 1405.04 nm, while the pore area was variable for the membranes made with the complete extract, ranging from 2139.21 nm in DA + PVP membranes to 3132.52 nm in OFI + PVP membranes, increasing the diameter of the pores up to 200%. Otherwise, the pore of TP + PVP membranes had a diameter of 1361.8 nm, barely decreasing 4%. In the case of membranes manufactured with phenolic fractions, the average pore area of the membranes was reduced, with measures of FR-DA + PVP: 152.6 nm, FR-OFI + PVP: 172.18 nm, and FR-TP + PVP: 181.88 nm, decreasing the pore size by 88% compared to that of the PVP membranes, as can be seen in Figure 3.

The fractionation of the compounds allows us to observe that the phenolic compounds participate as variables in the polymerization of the nanofibers, decreasing their diameter in their presence, which is also proportionally related to the pore size. We believe that during the spinning of the NFs, they do so in a similar way to ropes, winding fibers around themselves to increase their diameter, so in the presence of phenolic compounds, modifications occur, both in the medium and in the fibers, that limit the interaction between them, preventing the polymerization of larger-diameter fibers, which, in turn, favors an apparent increase in the number of fibers, in this way decreasing the size of the pores in the membranes.

Finally, Figure 4 shows the size distribution of nanoparticles present in the membranes made with the extract. We believe that the formation of structures on the fibers is due to the nature of the compounds present in the polymerization media, being those present in the TP extract with a greater affinity between them, agglomerating between them in initial stages within the polymerization medium without interfering in the lattice structure of the fibers, while for the DA and OFI extracts, the later formation of these agglomerates allows the interaction of the nanoparticles with the functionalized fibers, anchoring on its surface and obstructing direct interaction with each other, generating pores of greater area. The summary of all the data described here is found in Table 2.

### 3.2. Structural Characterization

After the manufacture of the membrane, these were characterized by FTIR spectroscopy. Although there are spectra of the extracts obtained from each plant and the phenolic fractions, these simply serve as a reference to know the reactive functional groups of the compounds present but not to identify them due to the great variety of them present, even after fractionation.

The spectra of the plant extracts are presented in Figure 5a. We could identify three main regions: one region from 3700 cm^−1^ to 2800 cm^−1^, related to the presence of amines and alcohols; a second region from 1900 cm^−1^ to 1000 cm^−1^, where, in turn, we can observe three peaks in all spectra, the first being approximately from 1990 cm^−1^ to 1500 cm^−1^, attributable to the large presence of compounds with carbon–oxygen bonds such as esters, carboxylic acids, amines and ketones; the second peak, from 1500 cm^−1^ to 1200 cm^−1^, referring to the attributable carbon–carbon bond tension, mainly to alkanes, in addition to the presence of alcohols and aromatic compounds; and the last peak found was from 1200 cm^−1^ to 1000 cm^−1^, mainly attributable to esters and alcohols. Finally, in the third region, from 1000 cm^−1^ to 500 cm^−1^, we mainly found the identity signatures of the compounds attributable to the delocalization of hydrogens in their structure.

With respect to each plant, based on its spectrum, we can reach certain conclusions about the compounds present. DA presents a broader peak in the first region; however, a secondary peak at 3100 cm^−1^ stands out, which, in addition to corresponding to the presence of OH groups, can be related to the presence of aromatic compounds in the extract, also related to the presence of a broad band in region three, attributable to multiple delocalized H signals present in these compounds; however, the intensity in the spectrum, higher than in the other extracts, tells us of a wide variety and concentration with respect to the other plants. Likewise, the increase in the intensity of the second and third peaks in the second region is related to the presence of aromatic compounds. Finally, we can also find a peak at 3700 cm^−1^, attributable to free alcohols in the medium. While the OFI extract shows a composition similar to DA at a lower concentration, with different compounds attributable to the change in the morphology of the peaks in the third region, TP shows a slightly different presence of compounds, with a lower presence of compounds with OH groups, in addition to a lower presence of aromatic compounds due to the lower intensity of peak 3100 cm^−1^, simply appreciating small secondary peaks in the vicinity. The change in morphology in the first peak of the second region also stands out; the first head is mainly attributable to carboxylic acids and esters, while the second confirms the presence of aromatic rings.

After the fractionation of the extract into various partitions, it was spectrally characterized, as can be seen in Figure 5b, where we can see that the morphology of the spectra is apparently maintained, mainly by cleaning up the noise in them in addition to some specific changes from plant to plant. We can observe, in the first instance, the disappearance of the peak at 3700 cm^−1^, referring to livery alcohols, in addition to a better delimitation of the widest peak in the first region with a decrease in the secondary peak at 3100 cm^−1^, referring to aromatic compounds, and finally, the decrease in the intensity of the third peak of the second region as well as multiple signals present in the third region, which strongly suggests a high presence of esters, alcohols, carboxylic acids, and aldehydes in the extract, which, after fractionation, have been separated into other fractions. OFI maintains a similar pattern to DA, as mentioned above, with TP again being the one with the greatest changes, even with hardly any signals in the vicinity of 3100 cm^−1^ that are attributable to aromatic compounds, which suggests that the majority of phenolic compounds present less complexity than that present in other plants, in addition to the increase in intensity in the second peak of the second region, attributable to the OH groups present in the phenols, which by removing the other compounds allowed its signal to be found.

Additionally, in Figure 6, all the membranes were compared against the PVP membrane, revealing the most important differences present at 1750 cm^−1^, in the region between 3600 cm^−1^ and 3300 cm^−1^, and the increase in intensity in the region between 3000 cm^−1^ and 2800 cm^−1^, changes that could be attributed to the opening of the rings present in the molecules generating reactive seeds for the incorporation of other compounds, a process in which we believe the OH groups present in the medium for the phenols interact with the nitrogen–carbon bond of the PVP ring, breaking the ring and exposing the nitrogen, forming a secondary amine, while the carbon of this bond would remain protonated, being reduced by the OH in the medium to form a carboxylic acid. However, as we have mentioned, the concentration of these phenolic groups would mediate the reaction, so the higher the concentration of these compounds, the greater the number of reaction seeds generated, with the TP plant being the one with the greatest activity and the one with the greatest changes in its spectrum. The initial PVP molecule and the proposed product after the interaction and reaction with the groups present in the medium are shown in Figure 7. Similar changes in the spectrum and the PVP structure due to the addition of oxygen through photooxidation have been reported previously [37].

In addition, the FTIR spectra of the membranes conjugated with extracts presented another series of changes in bands within positions 731, 1018, 1101, 1117, 1250, and 1269 cm^−1^, indicating the conjugation of the compounds present in the extracts with the fibers of PVP due to the presence of the seed reaction sites, which allows interaction with other compounds; however, their own nature is what establishes the shape and size of the agglomerates present in the fiber, since TP fibers have a greater degree of surface functionalization, an undetermined specific compound is necessary to initiate this reaction, which seems to be found in higher concentration in the other extracts, which seem to be aromatic in nature, with alcohols and even phenols attributable to the present peaks.

## 4. Conclusions

In this study, electrospun PVP membranes were obtained with extracts and phenolic fractions of the plants *Dysphania ambrosioides*, *Opuntia ficus-indica*, and *Tradescantia pallida*. Likewise, the conditions for the fabrication allowed us to obtain results of greater importance that allow us to partially elucidate the steps that occur for the functionalization of the fibers and the incorporation of other compounds into them. The opening of the PVP rings not only allows the incorporation of other compounds as we can see in the membranes manufactured together with the extracts, which we originally pursued to generate therapeutic membranes, but it also allows the destabilization of the fibers, allowing a better degradation of them, avoiding their accumulation as microplastics. Likewise, the generation of reaction seeds on the fibers through functionalization would allow the addition of a greater diversity of compounds. Future studies under more controlled conditions should be performed to fully elucidate the reaction mechanisms and the participating compounds for their applications.

## Figures and Tables

**Figure 1 polymers-15-04720-f001:**
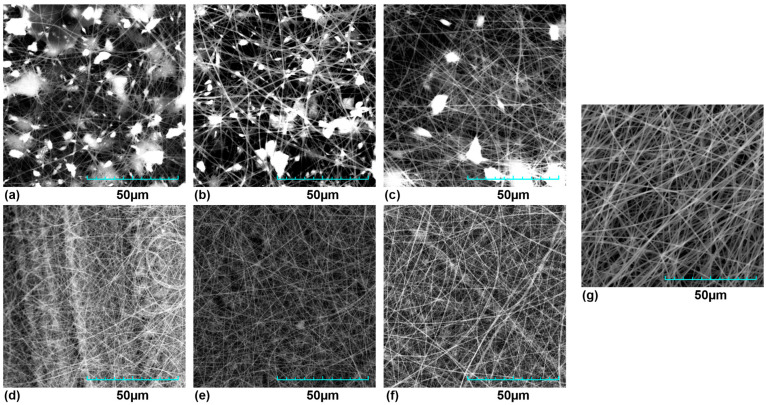
SEM micrograph of membranes. (**a**) DA + PVP, (**b**) OFI + PVP, (**c**) TP + PVP, (**d**) FR-DA + PVP, (**e**) FR-OFI + PVP, (**f**) FR-TP + PVP, and (**g**) PVP Control.

**Figure 2 polymers-15-04720-f002:**
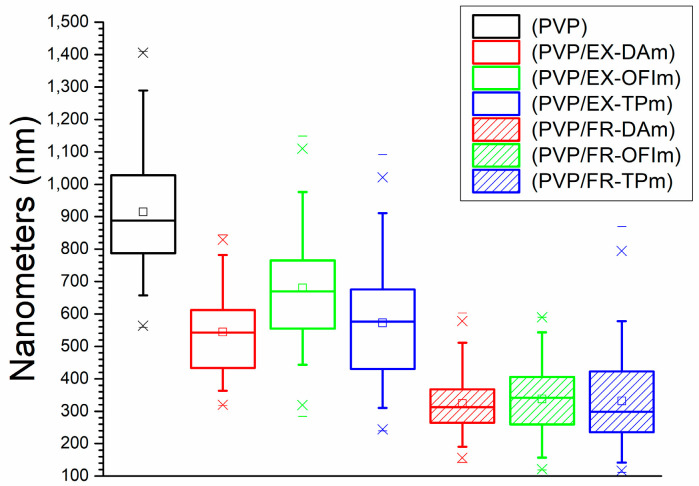
Diameter of the nanofibers.

**Figure 3 polymers-15-04720-f003:**
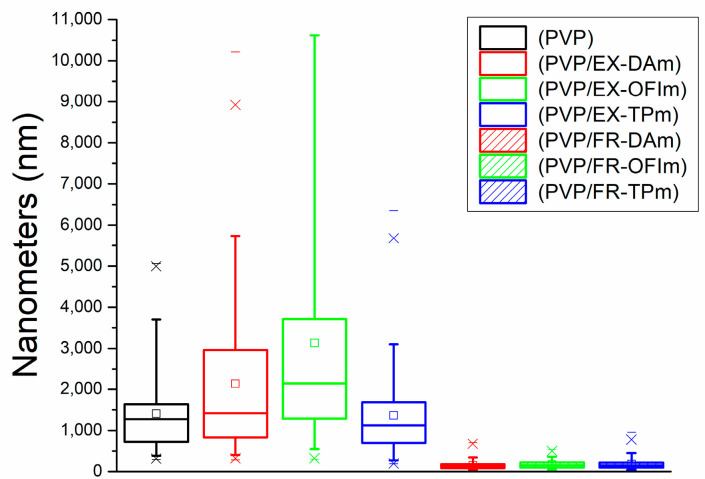
Porosity of the membranes.

**Figure 4 polymers-15-04720-f004:**
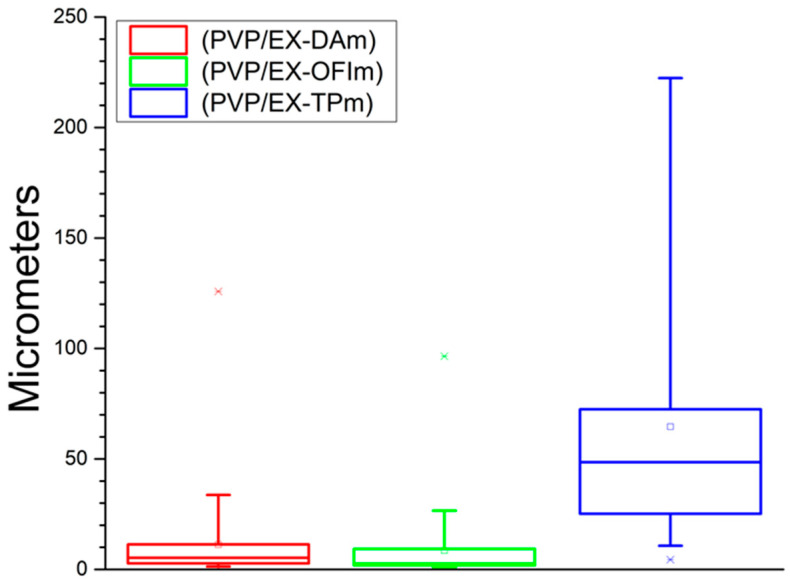
Diameter of the nanoparticles (rosaries).

**Figure 5 polymers-15-04720-f005:**
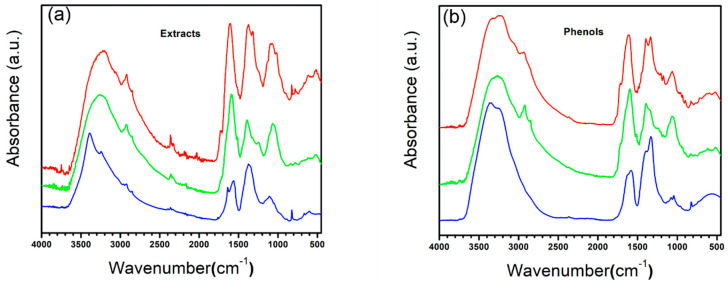
FTIR spectra: (**a**) extracts and (**b**) phenol fractions. (Red) DA, (Green) OFI, and (Blue) TP.

**Figure 6 polymers-15-04720-f006:**
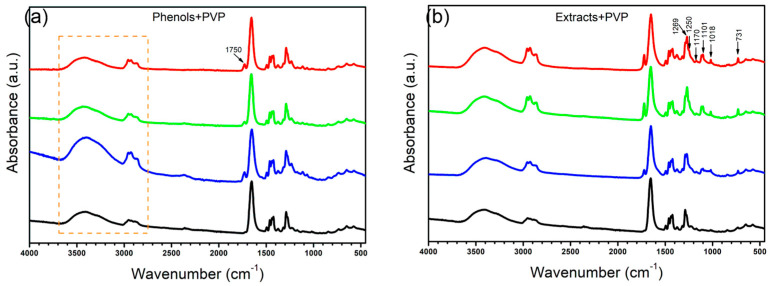
FTIR spectra of membranes. (**a**) Membranes made with complete extracts; (**b**) membranes made with the phenol fractions. (red) DA, (green) OFI, (blue) TP, (black) PVP control.

**Figure 7 polymers-15-04720-f007:**
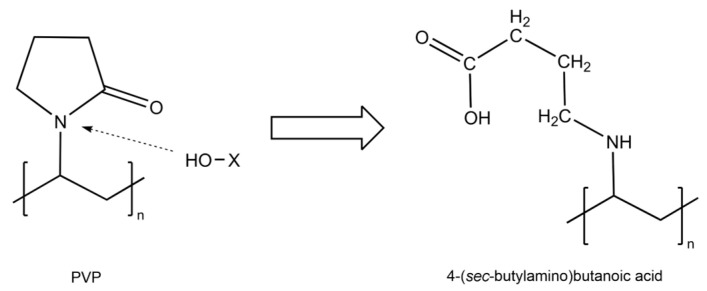
Proposed reaction mechanism for the interaction between phenolic compounds (HO-X) and PVP.

**Table 1 polymers-15-04720-t001:** Sample Labels.

Samples	Membranes
Polyvinylpyrrolidone	PVP
Extract (*Dysphania ambrosioides*) + Polyvinylpyrrolidone	DA + PVP
Extract (*Opuntia ficus-indica*) + Polyvinylpyrrolidone	OFI + PVP
Extract (*Tradescantia pallida*) + Polyvinylpyrrolidone	TP + PVP
Phenolic fraction (*Dysphania ambrosioides*) + Polyvinylpyrrolidone	FR-DA + PVP
Phenolic fraction (*Opuntia ficus-indica*) + Polyvinylpyrrolidone	FR-OFI + PVP
Phenolic fraction (*Tradescantia pallida*) + Polyvinylpyrrolidone	FR-TP + PVP

**Table 2 polymers-15-04720-t002:** Average data of morphological characterization of membranes.

Membranes	NFD ^1^	NFD SD ^1^	PA ^2^	PA SD ^2^	NPA ^3 +^	NPA SD ^3 +^
PVP	915.17	185.64	1405.04	997.84	-	-
DA + PVP	544.04	125.08	2139.21	18,116.57	11,261.54	19,083.83
OFI + PVP	680.50	166.38	3132.52	2983.64	8504.04	150,252
TP + PVP	572.57	172.85	1361.80	1043.10	64,588.08	65,905.26
FR-DA + PVP	323.12	92.32	152.60	110.09	-	-
FR-OFI + PVP	338.21	115.26	172.18	101.24	-	-
FR-TP + PVP	331.48	138.22	181.88	137.356	-	-

^1^ Average of nanofiber diameter (NFD), ^2^ pore area (PA), ^3^ nanoparticle area (NPA), Rosary (+), and standard deviation (SD). All the measurements are in nm.

## Data Availability

Data available on request to the corresponding author.

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
