# Peer review of "Generation of PVP Membranes Using Extracts/Phenolic Fraction of Dysphania ambrosioides, Opuntia ficus-indica, and Tradescantia pallida"

_polymers, 2023, doi:10.3390/polym15244720_

Round 1

Reviewer 1 Report

Comments and Suggestions for Authors

Electrospinning polyvinylpyrrolidone (PVP) nanofiber membranes were accomplished with
extracts and phenolic fractions of Dysphania ambrosioides (epazote), Opuntia ficus-indica
(nopal), and Tradescantia pallida (chicken grass) in the present study. Fourier transform infrared
spectroscopy and scanning electron microscopy were implemented to characterize the membrane
diameter and pore size.
Decision: Major Revision
1. Page 1: issue in keywords. Therapeutical is not used in the abstract and is mentioned in
keywords. Generally, the keywords were chosen from the abstract. Update the keywords.
2. Do not use abbreviations in keywords.
3. Page 2, Line 81: pulverized is not a scientific terminology. Kindly replace it with "reduce
to smaller nanoparticles."
4. Page 2, Line 85 Why the extract was stored at 4 °C, mention the reason. Why can’t they be
stored at room temperature?
5. Page 3, Line 98: Kindly provide the schematic of the preparation of membranes and also
show the instrument used for the preparation of membranes in the schematic.
6. In SEM images, can you provide the nanofiber images in the 200 nm range so one can
easily visualize the diameter reduction in different membranes after treating them with
various extracts?
7. Why is the diameter reduced less when in the case of (a) DA+PVP, (b) OFI+PVP, and (c)
TP+PVP and reduced significantly when phenol is included with it?
8. What are agglomerations and bright spots present in SEM images of (a) DA+PVP, (b)
OFI+PVP (c) TP+PVP.
9. Provide the evidence of Figure 2. The Provided SEM images did not signify the difference
in diameter of NF having different extracts. They almost show the same diameter.
10. How the porosity or pore diameter is evaluated as presented in Figure 3.
11. Provide SEM images of nanoparticles as mentioned in Figure 4.
12. Page 7, Line 194 Supplementary data is mentioned but not provided.
13. Page 7, Figure 5. The FT-IR analysis of Membranes made with complete extracts and the
Phenols fractions is almost similar. There is no major significance or difference between
them, only the one OFI (Blue) curve at a wavelength of 3500-3000 cm-1. Kindly plot them
together so one can visualize the difference between them.
14. Figure 5 Y-axis labeling issue. Kindly replace it with absorbance (a.u)

Author Response

We appreciate your comments and feedback, and we provide responses based on them below. On the other hand, the revised document is attached with the corresponding changes highlighted in blue, as well as other errors found in yellow.

Reviewer 2 Report

Comments and Suggestions for Authors

Generation of PVP membranes using extracts/phenolic fraction of Dysphania ambrosioides, Opuntia ficus-indica, and Trades cantia pallida is interesting.

The work presented here is exciting but requires major revisions before it can be accepted for publication. My comments are as follows:

1. Authors need to be more careful about referencing.

 (a) “There are many studies that affirm that “traditional” healing methods are useless, including increasing the probability of contracting some type of infection, delaying the healing process, causing discomfort, and more pain to the patient” Please add more than one reference.

(b) “In recent decades, active healing devices have been developed such as hydrogels, foams, hydrocolloids, and ointments which have helped promote the healing process.”- Please add references.

(c) “That is why various work groups around the world seek to overcome the challenges presented by the wound healing process by proposing new technologies and innovative devices that intervene actively and dynamically with the different elements and events that occur in the healing processes”- Please add more than one reference.

2. In Figure 1, the author can add the scale bar inside of the SEM images.

3. Please level all the curves of Figure 5 (Inside of the Figures)

4. Please provide the SEM images of different particle ranges (maybe 5 μm, 10 μm and 100 μm).

5. Have you tried the BET porosity analysis of the nanoparticle-based membranes?

6. Please identify the important peaks in the figure 5

Author Response

(The authors gave the same response as above.)

Round 2

Reviewer 1 Report

Comments and Suggestions for Authors

Accept in present form.

Reviewer 2 Report

Comments and Suggestions for Authors

Recommendation: Publish as is; no revisions needed.

Comments:

After carefully reading the revised manuscript and point-by-point response to reviewers' comments, I can fully understand the authors' argument and purpose. Thus, I recommend this paper for publication without further modification.